# Knowledge of prevention of COVID-19 among the general people in Bangladesh: A cross-sectional study in Rajshahi district

Md. Masud Rana[1], Md. Reazul Karim[1], Md. Abdul Wadood[2], Md. Mahbubul Kabir[3], Md. Mahidul Alam[4], Farhana Yeasmin[5], Md. Rafiqul Islam[6]*

1 Department of Statistics, University of Rajshahi, Rajshahi, Bangladesh, 2 Medical Center, University of Rajshahi, Rajshahi, Bangladesh, 3 Department of Management Studies, University of Rajshahi, Rajshahi, Bangladesh, 4 Department of Medicine, Rajshahi Medical College Hospital, Rajshahi, Bangladesh, 5 Department of Tourism and Hospitality Management, University of Dhaka, Dhaka, Bangladesh, 6 Department of Population Science and Human Resource Development, University of Rajshahi, Rajshahi, Bangladesh

* rafique_pops@yahoo.com

## Abstract

### Background

Until now, no vaccine or effective drug is available for the control, prevention, and treatment of COVID-19. Preventive measures are the only ways to be protected from the disease and knowledge of the people about the preventive measures is a vital matter.

### Objectives

The aim of the study was to assess the knowledge of the general people in Rajshahi district, Bangladesh regarding the COVID-19 preventive measures.

### Methodology

This cross sectional study was conducted from March 10 to April 25, 2020. Data were collected with a semi-structured questionnaire from 436 adult respondents selected by using a mixed sampling technique. Frequency analysis, chi-square test, and logistic regression model were utilized in this study. SPSS (IBM, Version 22) was used for data analysis. 95% confidence interval and p-value = 0.05 were accepted for statistical significance.

### Results

Only 21.6% of the respondents had good knowledge of the COVID-19 preventive measures. The highest 67.2% of them knew that washing hands with soap could prevent the disease, but contrarily, the highest 72.5% did not know that avoidance of touching mouth, nose, and eyes without washing hands was a preventive measure. Only 28.4% and 36.9% of the respondents knew that maintaining physical distancing and avoiding mass gatherings were measures of prevention of COVID-19 respectively. The younger age ($\leq$25 years), low family income ($\leq$15,000 Bangladeshi Taka (BDT), occupation others than business and service,

**Data Availability Statement:** All relevant data are within the paper and its Supporting Information files.

**Funding:** No fund for publication.

**Competing interests:** The authors declare that they have no competing interests.

and nuclear family had the lower odds of having no/less knowledge about the preventive measures.

## Conclusions

The knowledge level of the general people regarding prevention of COVID-19 was alarmingly low in Bangladesh. The government of Bangladesh, health policy makers and donor agencies should consider the findings and take immediate steps for improving knowledge of the public about prevention of the disease.

## Introduction

The coronavirus disease 2019 (COVID-19) is an acute respiratory disease caused by the 2019 Novel coronavirus (nCoV) that mainly affects the lungs [1]. The virus is transmitted by droplets, fomites and close contacts. The source and disease progression of this new virus are yet to be fully understood that necessitates preventive measures until effective treatment and vaccine are available [1]. The disease first broke out at Wuhan city of the Hubei Province in China [2]. It has quickly spread resulting in an epidemic across the country, followed by a pandemic with increasing number of cases in different countries throughout the world [3]. The World Health Organization (WHO) announced a public health emergency in late January 2020 and described it as a pandemic in March 2020 [4]. The pandemic has continued to cause escalating numbers of cases and death, and as of July 23, 2020, a total of 1, 53, 01,530 confirmed cases of COVID-19 and 6,25,005 deaths were recorded in more than 180 countries of the world [5]. WHO and national infection prevention guideline government of Bangladesh prescribed measures for prevention of COVID-19. The guidelines emphasized on (i) maintaining physical distancing (at least one meter), (ii) washing hands with soap for at least 20 seconds or using hand sanitizers, (iii) not touching nose, mouth, and eyes with unwashed hands, (iv) staying at home, (v) avoiding crowds, and so on [6–8].

The COVID-19 situation is aggravating every day in Bangladesh. As of July 23, 2020 a total of 2, 16,110 confirmed cases were identified and 2,801 deaths were recorded [9]. The government and many non-government organizations have come out, although in late, to fight the fatal disease. The whole country has been locked down. Almost all the government and non-government offices and educational institutions have been closed. All kinds of media are campaigning for creating awareness among the general people on prevention of the disease. But media are reporting that a considerable number of people are not following the government directives and preventive guidelines. One study has assessed the knowledge, attitude, practice and perception toward COVID-19 among students in Rajshahi University, Bangladesh [10]. Another studied knowledge and perception of Bangladeshi people towards COVID-19 [11]. Both the studies were conducted in the early stage of the COVID-19 outbreak in Bangladesh.

The COVID-19 pandemic has created a very high public health threat for the country. Considering the perspective, we aimed to assess the knowledge on COVID-19 prevention among people in Rajshahi district, Bangladesh.

## Methods

### Design and study population

This was a cross-sectional study. Our target area of study was Rajshahi district, Bangladesh. The district comprises of nine upazilas (sub districts) and a city corporation with 2,425.37sq.

km area. A total of 25, 95,197 people reside in the district which constituted our study population [12].

## Sample size determination

Since our target population is known (25, 95,197), the following formula was used for determining the sample size: $n = N/ (1+Nd^2)$, where n = required sample size, N = population size (25, 95,197), d = marginal error (0.05) [12, 13]. The formula provided that the minimum sample size of 400 would be sufficient for this study. Additionally, 36 data were collected for better outcome so, our final sample size was 436.

## Sampling

A mixed (probability and non-probability) sampling approach was adopted to select the samples. In the first stage, 3, out of 9 upazilas, and 1 out of 30 wards in the City were randomly selected by lottery. In the second stage, an average number of 100 people were selected purposefully from the selected upazillas and ward. But, during the survey in field level, 436 respondents were interviewed.

## Data collection

Data were collected from March 10[th] to April 25[th], 2020 using a semi-structured questionnaire. The following types of information were collected: (i) socio-demographic characteristics and (ii) knowledge on prevention of COVID-19 infection. Eight fully trained and experienced field researchers collected data from the respondents by face-to-face interview maintaining the COVID-19 preventive guidelines of WHO.

## Outcome variable

The outcome variable in our study is the knowledge on prevention of COVID-19. It was assessed based on answers of four components of one question. The question was: (a) Do you know how to prevent COVID-19? The answer of this question were (i) Maintaining physical distance, (ii) Avoiding mass gatherings, (iii) Hand washing for 20 seconds by soap, and using hand sanitizer (iv) Avoiding touch of mouth, nose and eyes. The respondents were divided into two groups based on their knowledge levels: (i) Knowledgeable- those who gave correct answers to all the four questions have been considered possessing good knowledge, (ii) Not knowledgeable- those giving 0–3 correct answers.

## Independent variables

We included theoretically pertinent socioeconomic and demographic factors as independent variables. In this study, we classified subject's age into two groups: ≤25 years and ≥26 years, gender into two groups: male and female, marital status as ever-married and unmarried, and occupational group was categorized into three groups such as business, service and others. Education was classified based on the formal education system in Bangladesh: primary education, secondary, and higher. Place of residence was categorized as rural and urban. Subject's family type was categorized as nuclear and joint. Respondent's monthly income was categorized as ≤15000 Bangladeshi Taka (BDT) or ≥15001 BDT.

## Statistical analyses

This study conducted frequency distribution of participants' answers to items for preventive measures of COVID-19, and the chi-square test was used to investigate the association of level

of knowledge among the socio-economic factors. Furthermore, binary logistic regression model was utilized to assess the effect of the associated factors on level of knowledge of preventive measured regarding COVID-19.

## Ethical approval

The ethical approval was received from Institute of Biological Science (IBSc) memo no: 64/320IAMEBBC/IBSC, University of Rajshahi, Bangladesh. We also received written consent from the subjects.

## Results

Table 1 shows the socio-economic and demographic profiles of the respondents. Of the total 436 respondents, approximately 68.3% were below ≤25 years of age and 67.7% were male. 65.4% were unmarried, and 67.4% respondents lived in the urban area. 48.6% of the participants passed primary education level, 24.1% secondary level, and the remaining 27.3% had higher level of education. According to occupation, 23.6% of them were businessmen, 35.8% service holders, and 40.6% had others occupation. 87.4% subjects came from the nuclear family and 50.7% respondents had monthly family income of ≤15,000 BDT.

Frequency of the respondents' answers to the four items of knowledge of COVID-19 preventive measures was shown in Table 2. Only 28.4% and 36.9% of the respondents knew that maintaining social distancing and avoiding mass gatherings were measures of prevention of COVID-19 respectively. A higher number of people (67.2%) had knowledge that washing hands with soap could prevent the disease, but contrarily, only 27.5% of the respondents knew avoiding touch of mouth, nose, and eyes without washing hands was a preventive measure.

Only 21.6% knew all the four measures of COVID-19 prevention (Fig 1).

In Table 3, association of socioeconomic and demographic factors and knowledge of prevention measures was shown. Age, occupation, type of family and monthly family income were found to have statistically significant association (p<0.001).

Statistically significant associated factors were considered as independent variables in the binary logistic regression analysis. The findings were presented in Table 4. It was demonstrated that the subjects of ≤25 years of age were less likely to have COVID-19 prevention

**Table 1. Socio economic and demographic profile of subjects.**

| Variables | N | (%) | Variables | N | (%) |
|---|---|---|---|---|---|
| **Age in years** | | | **Education** | | |
| ≤25 years | 298 | 68.3 | Primary | 212 | 48.6 |
| ≥26 years | 138 | 31.7 | Secondary | 105 | 24.1 |
| **Gender** | | | Higher | 119 | 27.3 |
| Male | 295 | 67.7 | **Occupation** | | |
| Female | 141 | 32.3 | Business | 103 | 23.6 |
| **Marital status** | | | Service | 156 | 35.8 |
| Unmarried | 285 | 65.4 | Others | 177 | 40.6 |
| Ever Married | 151 | 34.6 | **Type of family** | | |
| **Residence** | | | Nuclear | 381 | 87.4 |
| Urban | 294 | 67.4 | Joint | 55 | 12.6 |
| Rural | 142 | 32.6 | **Monthly family income** | | |
| | | | ≤15000 | 221 | 50.7 |
| | | | ≥15001 | 215 | 49.3 |

**Table 2. Frequency distribution of participants' answers to items of preventive measures.**

| Knowledge variables | Category | Number (n) | Percentage (%) |
|---|---|---|---|
| Maintaining social distancing. | No | 312 | 71.6 |
| | Yes | 124 | 28.4 |
| Avoiding mass gatherings. | No | 275 | 63.1 |
| | Yes | 161 | 36.9 |
| Washing hands with soap. | No | 143 | 32.8 |
| | Yes | 293 | 67.2 |
| Avoiding touch of mouth, nose, and eyes with unwashed hands. | No | 316 | 72.5 |
| | Yes | 120 | 27.5 |

knowledge than the subjects of ≥26 years of age (AOR = 0.301, 95% CI: 0.165–0.547; p<0.001). Businessmen and service holders had more likelihood of being knowledgeable than the respondents of others occupation (AOR = 8.586, 95% CI: 3.400–21.681; p<0.001) and (AOR = 14.048, 95% CI: 6.459–30.551; p<0.001) respectively. Nuclear family members were less likely to have knowledge on COVID-19 preventive measures compared to the joint family members (AOR = 0.099, 95% CI: 0.048–0.204; p<0.001). The respondents with monthly family income of ≤15,000 BDT showed less chance of having prevention knowledge of COVID-19 than the subjects with monthly family income of ≥15,001 BDT (AOR = 0.321, 95% CI: 0.176–0.587; p<0.001).

## Discussion

The present survey was conducted at the initial phase of the COVID-19 outbreak in Bangladesh. Mentionable that, by the time, the pandemic has already been a hot cake for all kinds of media, as well as the general people in the country. The government and many non-government organizations started huge campaigns on mode of transmission, signs and symptoms,

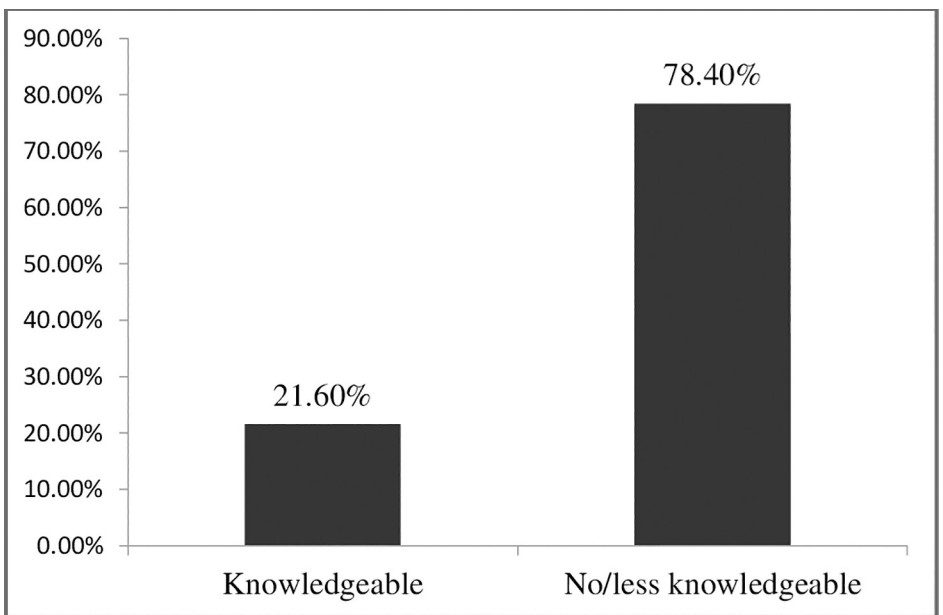

**Fig 1. Level of knowledge of preventive measures of COVID-19 among the general people in Rajshahi district, Bangladesh.**

**Table 3. Association between prevention knowledge level about COVID-19 and socio economic and demographic factors.**

| Variables | Knowledgeable (21.60%) | p-value | Variables | Knowledgeable (21.60%) | p-value |
|---|---|---|---|---|---|
| **Age** | | 0.001 | **Education** | | 0.267 |
| ≤25 years | 40(13.4) | | Primary | 45(21.2) | |
| ≥26 years | 54(39.1) | | Secondary | 18(17.1) | |
| **Gender** | | 0.197 | Higher | 31(26.1) | |
| Male | 69(23.4) | | **Residence** | | 0.400 |
| Female | 25(17.7) | | Urban | 60(20.4) | |
| **Marital status** | | 0.913 | Rural | 34(23.9) | |
| Unmarried | 61(21.4) | | **Type of family** | | 0.001 |
| Married | 33(21.9) | | Nuclear | 27(49.1) | |
| **Occupation** | | 0.001 | Joint | 67(17.6) | |
| Business | 29(28.2) | | **Monthly family income** | | 0.001 |
| Service | 62(39.7) | | ≤15,000 | 20(9.0) | |
| Others | 03(1.7) | | ≥15,001 | 74(34.4) | |

and preventive measures of COVID-19. Media were giving all up-to-date information regarding the pandemic in home and abroad. Rajshahi is the divisional head quarter, city and district, famous as educational, silk and clean city in Bangladesh, so this study assumed that the population would be knowledgeable but our study found very poor knowledge on COVID-19.

Repatriation of thousands of Bangladeshi people firstly from China in January and then from all other COVID-19 affected countries, and the mismanagement of their screening, quarantine, and isolation created a huge debate both in media and the public [14]. All our respondents were educated and most of them came from urban settings. So, it was assumed that they would have sufficient knowledge of prevention of COVID-19. But our findings revealed that, unexpectedly, only 21.6% of them were knowledgeable. A recent study got similar type of findings that Bangladeshi people had poor knowledge (8.56±1.63, on a scale of 13.0) with about 50% knowing that the use of surgical masks was a way of prevention of COVID-19. Similar results were found in Saudi Arabia [15]. However, a Bangladeshi study found that 72.2% of people including a good lots of doctors and medical students knew hand hygiene, covering nose and mouth while coughing, and avoiding sick contacts could help prevention of COVID-19 transmission [10]. The knowledge level of our respondents was much lower than that of

**Table 4. Effect on level of prevention knowledge on COVID-19 and socio economic and demographic factors.**

| Variables | B | S.E. | P-value | Adjusted Odds Ratio (AOR) | 95% CI for AOR | |
|---|---|---|---|---|---|---|
| | | | | | Lower | Upper |
| **Age in years** | | | | | | |
| ≤25 vs ≥26 [R] | -1.201 | 0.305 | 0.001 | 0.301 | 0.165 | 0.547 |
| **Occupation** | | | | | | |
| Business vs Others [R] | 2.150 | 0.473 | 0.001 | 8.586 | 3.400 | 21.681 |
| Service vs Others [R] | 2.642 | 0.396 | 0.001 | 14.048 | 6.459 | 30.551 |
| **Type of family** | | | | | | |
| Nuclear vs Joint [R] | -2.316 | 0.371 | 0.001 | 0.099 | 0.048 | 0.204 |
| **Monthly family income** | | | | | | |
| ≤15,000 vs ≥15,001 [R] | -1.136 | 0.308 | 0.001 | 0.321 | 0.176 | 0.587 |

**N. B.:** B-Coefficient; S. E-Standard Error; AOR- Adjusted Odds Ratio; CI- Confidence Interval: R-Reference

Chinese general people [16, 17]. In Nigeria, a study found 94.25% of people knew that regular hand washing and social distancing could prevent COVID-19 infection [18]. Knowledge level of the Egyptians were also found higher– proper hand-washing (99.6%), maintaining appropriate physical distance (99.1%), and avoiding touching eyes, nose, and mouth (97.1%) [19]. Bangladeshi people are usually less conscious about health and their practice level of good health measures is significantly poor. As for example, only 40% of people wash hands with soap and water in Bangladesh [19]. This unawareness of the general people might contribute to their poor attention to campaign on COVID-19 preventive measures ultimately resulting in their inadequate knowledge on that subject. The time of the survey might also be a factor. As the rate of infections and death from COVID-19 was still alarming, the general people might not seriously think about its grave impact and the preventive measures in spite of massive campaigns.

People of low-income family are supposed to be less conscious about health and the practice of health measures is usually insufficient among them. These factors might contribute to low level of knowledge on preventive measures for COVID-19 among our respondents. The reason behind high level of awareness among older adults ($\geq$26 years) might be that they gather knowledge about the threats and risks as they are more resolute about their own and family members' life. Businessmen and service holders usually meet different types of people every day and come to know current news regularly and repeatedly. This might help them gather high level of COVID-19-related perception. Egyptian respondents from low monthly income families had significantly lower level of knowledge which is consistent with our findings [19]. A Chinese study shows that older adults (30–49 years) and people doing mental labor had higher odds of getting more knowledge than the younger adults (16–29 years) and students and unemployed respectively [17].

As many days have passed in the meantime after our survey, and the COVID-19 pandemic is reaching its peak, the knowledge level of the public might change. Further study is needed.

### Strengths

The strength of this research is that it is the first study to analyze the knowledge of COVID-19 among the general people in Rajshahi district Bangladesh. The standardized questionnaire format was carefully developed to ascertain accurate information from the subjects. The interviewers were trained up, and the field works was monitored during the survey by the principal authors of this study. The data contained a low proportion of missing information. The study can be generalized to other areas in Bangladesh because similar characteristics exist all over the country. This study also identified some factors regarding knowledge on COVID-19 prevention through some sophisticated statistical tools and techniques.

### Limitations

We had some limitations. Due to lockdown situation, we could not follow strict random sampling technique. The sample size was small and not nationally representative. For the study design (cross sectional study), we could not look into any change in the people's level of knowledge on COVID-19 preventive measures in course of time. We did not consider all the components of preventive measures of COVID-19. Further studies are needed.

## Conclusions

The objective of this study was to assess the knowledge of preventive measures for COVID-19 among the general people in Rajshahi district, Bangladesh. We found that only 21.6% of people had good knowledge regarding COVID-19. The government of Bangladesh, Health Policy

makers and donor agencies should consider these findings while promulgating and implementing principles and guidelines for control and prevention of COVID-19 in Bangladesh

## Supporting information

**S1 File.**
(SAV)

## Acknowledgments

The authors gratefully acknowledge the Civil Surgeon Office Rajshahi District, Bangladesh for giving permission to take data from CCs catchment area. We also acknowledge Health Research Group, Department of Statistics, and University of Rajshahi, Bangladesh to help in data collection from CCs, and Our heartfelt thanks to Minara Jesmin, Graduate teaching partner, Hewens primary school. Hayes, UB4 8JP, London, UK for helping grammatical correction of this study.

## Author Contributions

**Conceptualization:** Md. Masud Rana, Md. Abdul Wadood, Md. Rafiqul Islam.

**Data curation:** Md. Masud Rana, Md. Mahidul Alam, Farhana Yeasmin.

**Formal analysis:** Md. Masud Rana.

**Methodology:** Md. Rafiqul Islam.

**Software:** Md. Mahidul Alam.

**Validation:** Md. Abdul Wadood.

**Writing – original draft:** Md. Masud Rana.

**Writing – review & editing:** Md. Reazul Karim, Md. Abdul Wadood, Md. Mahbubul Kabir, Md. Rafiqul Islam.

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
