## [Decision Letter · Decision Letter 0]

13 Sep 2020

PONE-D-20-23073

Knowledge of prevention of COVID-19 among the general people in Bangladesh: A cross-sectional survey in Rajshahi district

PLOS ONE

Dear Dr. Rafiqul Islam,

Thank you for submitting your manuscript to PLOS ONE. After careful consideration, we feel that it has merit but does not fully meet PLOS ONE’s publication criteria as it currently stands. Therefore, we invite you to submit a revised version of the manuscript that addresses the points raised during the review process.

We look forward to receiving your revised manuscript.

Kind regards,

Francesco Di Gennaro

Academic Editor

PLOS ONE

Additional Editor Comments:

Dear authors,

Follow reviewer suggestions to improve your article.

Journal Requirements:

Reviewers' comments:

Reviewer's Responses to Questions

**Comments to the Author**

1. Is the manuscript technically sound, and do the data support the conclusions?

Reviewer #1: Partly

Reviewer #2: Yes

2. Has the statistical analysis been performed appropriately and rigorously? 

Reviewer #1: No

Reviewer #2: Yes

3. Have the authors made all data underlying the findings in their manuscript fully available?

Reviewer #1: No

Reviewer #2: Yes

4. Is the manuscript presented in an intelligible fashion and written in standard English?

Reviewer #1: No

Reviewer #2: Yes

5. Review Comments to the Author

Reviewer #1: The topic for this study is important especially in the middle of a pandemic such as the world is experiencing now. The objective of the study is equally made clear to the reader. A number of issues were noticed as the manuscript was thoroughly read. The first has to do with the grammatical errors throughout the manuscript which need to be addressed. In the methods section, the sample size determination is confusing to readers. You talked about piloting study and combining that with calculation of the sample size. This is not clear. Are you stating that your sample size included those included in the pilot study? If that is the case, it may unusual since participants in pilot studies are usual part of actual study subjects. This could have introduced bias and negatively affect content and construct validity of the study. Again, there is no description of how the pilot study was conducted.

Although you have presented the frequency of age of the participants, your categorization of the age into just 25 years and under and 26 years and above is not comprehensive enough to show the diversity of the age of participants. Age is usually grouped in either five or ten year intervals so that readers get a clear picture of the age diversity of the participants.

In knowledge assessment, you used just four knowledge variables which is not comprehensive enough to measure knowledge of COVID-19 even though the four variables are basically what are in the public domain. The background of COVID-19 was not assessed except the four basic preventive methods. The conclusion drawn based on this study could be guessed by lay persons considering the time you conducted the data collection for this study. The entire world and the scientific world itself struggled to understand COVID-19 in terms of etiology, accurate modes of spread as demonstrated in different measures recommended by different medical personnel in different countries.

This study will be more meaningful if conducted today as there is now more comprehensive information and evidence from around the world to measure comprehensive knowledge of COVID-19.

Reviewer #2: This cross-sectional study was undertaken to assess the knowledge of COVID-19 preventive measures among adults in Bangladesh.

The following suggestion may help the Authors in improving the manuscript.

Abstract

Background: the sentence “…of the people about the preventive measures is a vital issue”. Can omit “issue”

Better to mention in the abstract that enrolled participants were adults.

Results: not clear what “good” knowledge means

“Physical” distancing is a better term than “social” distancing

Introduction

Line 64-69: cite reference

Line 66: should it be washing hands with soap for at least 20 seconds or use hand sanitizer?

Line 76: only two COVID-19-76 related published articles from Bangladesh? Is so, say so.

Line 78: better to mention “another study” instead of “another article”

Line 81: Which situation? I assume the COVID-19 pandemic? Say so.

Line 82: instead of “corona virus” mention “COVID-19”

Methods

Line 102, 107: instead of “corona virus” mention “COVID-19”

Line 108-109: use of hand sanitizer was not part of the questions asked?

Line 111: I think “Not Knowledgeable” suits better instead of “No Knowledgeable”

Results

Characteristics of the respondents: avoid duplication of information in the text if presented in the Table.

Line 149-150: Better to avoid using “proper” and “complete”. Better to say, Only 21.6% knew all the four measures of COVID-19 prevention.

Effect of associated factors on knowledge of preventive measures: avoid duplication of information in the text if presented in the Table.

Discussion

This section needs to be organized better: can begin with the summary of main findings, followed by other findings, evidence in context, strengths, and limitations

Conclusion

Avoid duplication of information presented in the Results.

6. PLOS authors have the option to publish the peer review history of their article (what does this mean?). If published, this will include your full peer review and any attached files.

Reviewer #1: No

Reviewer #2: No

---

## [Author Response · Author response to Decision Letter 0]

28 Oct 2020

Response to editor and reviewer comments

Response to editor comments/ Journals requirements:

1. It was written due to journal style.

2. All authors checked the manuscript thoroughly. Moreover, it was edited by Minara Jesmin, Graduate Teaching Partner, Hewens Primary School. Hayes,UB4 8JP, London, United Kingdom for helping grammatical correction of this study,

3. Data is available in the PLOS ONE journal website.

Response to reviewer’s comments:

1. Conclusions is revised as per reviewers-1 advice.

2. The statistical analysis has been used appropriately and rigorously according to the suggestions of reviewesrs-1.

3. Yes, data is available in the PLOS ONE journal website due to advice of reviewers-1.

4. Typographical or grammatical errors were corrected at revision by authors and Minara Jesmin, Graduate Teaching Partner, Hewens Primary School. Hayes,UB4 8JP, London, United Kingdom.

Reviewer comment’s to the authors:

Reviewer#1: The grammatical errors throughout the manuscript were corrected. The method sample size determination is revised. The word piloting omitted from the manuscript. It was noted that it was unfortunately stayed in the manuscript. Age is categories for better finding in the analysis of chi-square test and logistic regression analysis. Further study we will follow your advice. The knowledge on prevention of COVID-19 was assessed based on four components according to WHO and DGHS, Ministry of Health and Family Welfare, Government of Bangladesh. The conclusion is revised as per your mentorship. In future, we try to comprehensive study on COVID-19.

Reviewer#2: 

In Abstract: 

In Background: Issue is omitted

In Methodology: We used adult respondents

In methodology: Good knowledge indicates the respondents who answered four question yes.

In results: We write physical distancing instead of social distancing 

In introduction: 

Line 64-69: reference is cited

Line 66: The sentence is rewritten

Line76: This sentence is deleted

Line 78: We write another study instead of another article

Line81: The sentence is corrected 

Line 82: We write COVID-19 instead of corona virus

In Methods: 

Line 102,107: We write COVID-19 instead of corona virus

Line 108-109: The sentence is rewritten

Line111: We write not knowledgeable instead of no knowledgeable

In Results:

Duplication is removed

Line 149-150: The sentence is revised

Duplication is omitted

In Discussion:

The discussion section is rewritten due to suggestions of reviewer’s comments

Strengths and limitations are added in the manuscript

In conclusion:

The conclusion is revised according to reviewer’s suggestions

---

## [Decision Letter · Decision Letter 1]

23 Nov 2020

Knowledge of prevention of COVID-19 among the general people in Bangladesh: A cross-sectional survey in Rajshahi district

PONE-D-20-23073R1

Dear Dr.Md. Rafiqul Islam

We’re pleased to inform you that your manuscript has been judged scientifically suitable for publication and will be formally accepted for publication once it meets all outstanding technical requirements.

Kind regards,

Francesco Di Gennaro

Academic Editor

PLOS ONE

Additional Editor Comments (optional):

dear authors congratulations for your great paper

Reviewers' comments:

Reviewer's Responses to Questions

**Comments to the Author**

1. If the authors have adequately addressed your comments raised in a previous round of review and you feel that this manuscript is now acceptable for publication, you may indicate that here to bypass the “Comments to the Author” section, enter your conflict of interest statement in the “Confidential to Editor” section, and submit your "Accept" recommendation.

Reviewer #1: All comments have been addressed

Reviewer #2: All comments have been addressed

2. Is the manuscript technically sound, and do the data support the conclusions?

Reviewer #1: Yes

Reviewer #2: Yes

3. Has the statistical analysis been performed appropriately and rigorously? 

Reviewer #1: Yes

Reviewer #2: Yes

4. Have the authors made all data underlying the findings in their manuscript fully available?

Reviewer #1: Yes

Reviewer #2: Yes

5. Is the manuscript presented in an intelligible fashion and written in standard English?

Reviewer #1: Yes

Reviewer #2: Yes

6. Review Comments to the Author

Reviewer #1: I have read the revised manuscript several times in my review process and found that the authors have worked hard and thoroughly improved the manuscript. All the concerns I raised have been adequately addressed.

Reviewer #2: The authors have addressed comments made by the reviewers. However, there could still be some typos (eg. COVID-9.)

7. PLOS authors have the option to publish the peer review history of their article (what does this mean?). If published, this will include your full peer review and any attached files.

Reviewer #1: No

Reviewer #2: No

---

## [Editor Report · Acceptance letter]

1 Dec 2020

PONE-D-20-23073R1 

Knowledge of prevention of COVID-19 among the general people in Bangladesh: A cross-sectional study in Rajshahi district 

Dear Dr. Islam:

I'm pleased to inform you that your manuscript has been deemed suitable for publication in PLOS ONE. Congratulations! Your manuscript is now with our production department. 

Kind regards, 

on behalf of

Dr. Francesco Di Gennaro 

Academic Editor

PLOS ONE